# Manual Muscle Testing—Force Profiles and Their Reproducibility

**DOI:** 10.3390/diagnostics10120996

**Published:** 2020-11-25

**Authors:** Frank N. Bittmann, Silas Dech, Markus Aehle, Laura V. Schaefer

**Affiliations:** Division Regulative Physiology and Prevention, Department Sports and Health Sciences, University of Potsdam, 14476 Potsdam, Germany; bittmann@uni-potsdam.de (F.N.B.); dech@uni-potsdam.de (S.D.); aehle@uni-potsdam.de (M.A.)

**Keywords:** manual muscle testing, neuromuscular diagnostics, force profiles, reproducibility, adaptive force, handheld device

## Abstract

The manual muscle test (MMT) is a flexible diagnostic tool, which is used in many disciplines, applied in several ways. The main problem is the subjectivity of the test. The MMT in the version of a “break test” depends on the tester’s force rise and the patient’s ability to resist the applied force. As a first step, the investigation of the reproducibility of the testers’ force profile is required for valid application. The study examined the force profiles of *n* = 29 testers (*n* = 9 experiences (Exp), *n* = 8 little experienced (LitExp), *n* = 12 beginners (Beg)). The testers performed 10 MMTs according to the test of hip flexors, but against a fixed leg to exclude the patient’s reaction. A handheld device recorded the temporal course of the applied force. The results show significant differences between Exp and Beg concerning the starting force (p_adj_ = 0.029), the ratio of starting to maximum force (p_adj_ = 0.005) and the normalized mean Euclidean distances between the 10 trials (p_adj_ = 0.015). The slope is significantly higher in Exp vs. LitExp (*p* = 0.006) and Beg (*p* = 0.005). The results also indicate that experienced testers show inter-tester differences and partly even a low intra-tester reproducibility. This highlights the necessity of an objective MMT-assessment. Furthermore, an agreement on a standardized force profile is required. A suggestion for this is given.

## 1. Introduction

Manual Muscle Testing (MMT) is a widespread diagnostic tool all over the world. There is a broad variety of applications, e.g., in neurology, intensive care, physical therapy, osteopathy, sports medicine, and others. MMT is applied under many different intentions. The primary intention about 100 years ago was to determine the grade of muscle weakness caused by neurological issues like juvenile poliomyelitis [1]. Neurological disorders nowadays remain as one of the most important fields of application, especially in palsy caused by neurodegenerative illness. Other applications can be seen in rheumatological diseases as, e.g., dermatomyositis [2]. One of the most insufficiently understood fields are different forms of fatigue syndromes. They are regularly characterized, inter alia, by muscular weakness. Neurological examinations have revealed weakness in 58% of fibromyalgia patients (compared to 2% in healthy controls) [3]. Fatigue syndromes are also observed “for six months or more after clinical infection with several different viral and non-viral micro-organisms.” [4]. Those post-infectious fatigue syndromes are discussed as a possible pathway to chronic fatigue syndrome [4], which is also characterized by impairment of muscle function [4,5]. Against this background, muscle weakness after severe diseases of the respiratory system (and others) like COVID-19 [6] and especially intensive care are of high topicality. This ICU-acquired weakness (ICUAW) appears “as a secondary disorder while patients are being treated for other life-threatening conditions.” [7]. It is assumed as a result of myogenic or neurogenic dysfunctions, however, still without a plausible etiological idea [7]. The prevalence ranges between about 25 to 75% of the ICU-cases, depending on numerous factors [8]. Furthermore, muscle function seem to be affected in cancer [9], sarcopenia [10,11] or hormonal dysfunction [12,13]. Presumably, the largest field of application of MMT is the care of benign disturbances like orthopedic conditions. Therefore, MMT is a common tool applied by professional groups like physiotherapists, chiropractors, osteopaths, naturopaths, athletic training professionals and others.

Muscle function can be objectified by biomechanical measurements. One of the most common methods in rehabilitation is the measurement of the maximal strength using isokinetic dynamometry, which provides high objectivity and reproducibility [14]. Beyond the fact that this method is expensive, a disadvantage under clinical circumstances is the high expenditure of time required to carry out the measurements. In contrast, MMT can be done very quickly and is flexible for a great number of muscles within a short time. This has made it an essential tool in medical hands-on professions.

MMT can be performed in numerous ways. For clinical and scientific applications, it is usually defined how MMT must be executed. For instance, the Manual Muscle testing procedures of the National Institute of Environmental Health Sciences [2] prescribes the procedure of MMT in patients with juvenile myositis regarding the tested muscle and testing position, the order of testing, the muscle grading and even the exact wording of the commands [15]. The assessment of the neuromuscular function ranges from paralytic and paretic lesions (in neurological or myopathological cases) up to full power in healthy people. The results can be differentiated into 11 degrees [15]. In the lowest degrees, only movements of tendons or muscles are observed. In the upper degrees, the focus lies on the ability to hold against an external pressure applied by the examiner. Concerning the applied force (here called pressure) the authors define three different levels: slight, moderate, and strong, without further explanation of what this means. The manual also refers to the fact that the results of the upper grades “are heavily influenced by the stature of the subject and tester” [15]. To ensure a good reliability it is recommended to do backup MMTs by a second examiner with a similar stature.

Kendall et al. suggested a 6-degree scale ranging from palsy (“Gone—no contraction felt” [16]) up to full power (“normal—muscle can hold the test position against strong pressure” [16]). This scaling seems to be in accordance with Janda who also supposed a 6-degree rating scale, ranging from “no evidence of contractility” (Grade 0) to “normal” (Grade 5), whereby Grade 5 is characterized by a “very strong muscle with a full range of movement and able to overcome a considerable resistance.” [17]. Both approaches differentiate between a muscle strength and a muscle length test [16,17]. But in contrast to Kendall et al., Janda’s approach—based on the works of Daniels, Williams and Worthingham [18]—is aimed on a concentric contraction of the tested muscle (partly against external resistance provided by the examiner). Because the present study is focused on the so-called “break test” (see below), we further refer to Kendall et al. The muscle strength test includes the determination of the patient’s “strength of the muscle holding in the test position (…) against the examiner’s pressure” [16]. This is in accordance with the MMT after Goodheart, as it is utilized in Applied Kinesiology, which simply distinguishes between two states, a stable one (comparable with degree “normal” after Kendall et al. [16] or rather degree 10 appropriate to the IMACS manual [15]) and a state of instability (comparable with degrees “gone” or 10, respectively).

Comparing different MMT techniques—referring to the “strength test” of Kendall [16] or the MMT of Goodheart [19]—Conable and Rosner gave an overview of the kind of action between subject and tester [20]. During the so called “make test” the subject pushes against a fixed resistance offered by the tester. In contrary, the “break test” runs conversely. In this case, the subject has to resist a varying pushing action delivered by the tester [20]. Moreover, there are specific styles commonly used within particular communities, like, for instance, a patient started test, which begins with the patient pushing firstly but then merging into a “break test” [20].

This publication is in particular aimed at the “break test” procedure, which seems to be much more common than the “make test”. Its key element is the equilibrium between the isometric muscle actions of both the patient and the tester. As long as the interacting limbs stay in a stable position, both partners produce exactly the same isometric force. However, there are two crucial differences between them. Firstly, the tester acts in a pushing way, whereas the patient has to hold against it. So far, in motor science there is no distinction between a holding isometric muscle action (HIMA) and a pushing isometric muscle action (PIMA). This issue was rarely pursued in research and there is little scientific discussion on this topic. Electromyography (EMG) based studies found no differences and came to the point to reject a hypothesis of two different isometric actions [21,22]. Nevertheless, there are some hints which support the hypothesis. Some research showed a higher endurance in maintaining a submaximal pushing isometric action compared to a holding one, despite the identical applied force [23,24,25,26,27,28,29]. In addition, Hunter et al. [26] found differences in EMG at exhaustion and Rudroff et al. [23] revealed a distinction analyzing the EMG power frequency spectrum. Our own research analyzing the power spectrum of mechanomyographic signals (MMG) and the duration of maintaining a submaximal force level during HIMA or PIMA, respectively, delivered additional hints supporting the hypothetical idea of two different kinds of isometric muscle action [28,29]. Consequently, we assume that two different physiological mechanisms of isometric muscle action exist and are applied during the MMT by acting against each other.

Secondly, it is up to the tester to vary his or her pushing force (usually increasing it), but the patient has to react to it. The latter requires a complex regulative process of adaptation, including proprioception and sensorimotor control. Because the main task of the patient is to adapt to an applied external force we called this specific neuromuscular function the Adaptive Force (AF) [30,31]. The involved complex motor control network includes inter alia the motor cortex, thalamus, cerebellum, inferior olivary nucleus (ION), basal ganglia, cingulate cortex and the red nucleus [32,33,34,35,36,37,38,39,40,41,42,43,44,45,46,47,48,49,50,51,52,53,54,55,56]. In the process of adapting to external forces, this network has to deal with a time problem. The corrective response of an adaptive force on altering external forces needs about 80–100 ms [32,57], which is the long loop runtime through the circuits [58,59]. Therefore, in case of a varying external force input the adapting system has to anticipate the prospective change. Some research indicates that in particular a circuit between the cerebellum and the ION is involved in this forward controlling process [42,48,50,51,60,61]. This circuitry seems to be of special relevance for the characteristics of force profile applied by the tester during the MMT and will be discussed.

As mentioned above, quickness and flexibility are the advantages of the MMT in clinical practice. From a scientific point of view, the value of a diagnostic tool is characterized by objectivity and reproducibility. Because of the subjective nature of the MMT, the assessment and a discussion of these scientific quality criteria are urgently needed, although the review of Cuthbert and Goodheart (2007) concluded that studies speak for a good reliability and validity of the MMT [62]. There are two major aspects of subjectivity during the MMT depending on the tester: the temporal course of force development and the tester’s judgement concerning the patient’s response. The subjective judgement of the tester is based on a perception of the resistance offered by the patient. It ranges from “full resistance” or a kind of “locking” to “weakness” or a clear “yielding” (hereafter referred to as “stable” vs. “unstable”). The applied pressure is differentiated into the levels “slight”, “moderate” or “strong” [20]. The temporal profile of the force application (amount and time course) mostly remains unclear.

This paper does not deal with the aspect of judgement of the test-outcome by the tester (this will be addressed elsewhere). This work is focused on the reproducibility of the temporal force profile applied by the tester. Conable et al. [63] concluded that differences in magnitude and time course of force during the MMT may lead to different outcomes at the same patient and muscle. This point results in two crucial conclusions. Firstly, an appropriate intra-tester reliability will be a prerequisite to compare MMT outcomes along the timeline. This is what is mostly done in clinical practice. Secondly, the comparison of the MMT results between different testers requires matching testing techniques regarding the temporal course of force. Therefore, two main questions arise for the present study:Are the testers able to produce reliable force profiles (Intra-tester-reliability)?How comparable are the force profiles of different testers regarding their experiences?

To investigate these research questions, the force profiles of testers were measured excluding the reaction of the patient by fixing the patient’s limb. The investigation will lead to a suggestion of a standardized force profile based on neurophysiological considerations, which—of course—is open for discussion.

## 2. Materials and Methods

The investigation took place at the Neuromechanics laboratory of the University of Potsdam (Potsdam, Germany).

### 2.1. Participants

In total, 29 Caucasian healthy subjects, in the following called testers, participated in the study. The testers were recruited from graduates and lecturers of the training program “Applied Kinesiology based Integrative Medicine” of the Academy of Health and Exercise Therapy at the Brandenburg association of Health Promotion (Brandenburgischer Verein für Gesundheitsförderung e.V., BVfG, Potsdam, Germany) as well as from students of the University of Potsdam (study program M.Sc. Integrative Sports, Movement and Health Sciences). According to their self-reported MMT experience, they were classified into three groups:Experienced (Exp): more than four years of test experience and regular use of MMT in therapy practice. (*n* = 9)Little experienced (LitExp): At least the basic course of “Applied Kinesiology based Integrative Medicine” at the Academy of Health and Exercise Therapy at the BVfG; seldom use of MMT or less than 1 year regularly use. (*n* = 8)Beginners (Beg): no experience with MMT, partly a short first contact with the MMT during study programs at the University of Potsdam (BA Sports Therapy and Prevention or M.Sc. Integrative Sports, Movement and Health Science); no use of MMT. (*n* = 12)

The anthropometric data and test experiences are displayed in Table 1. Initially, it was planned to measure further experienced testers at the congress of the German Doctors society for Applied Kinesiology (DÄGAK) in September 2020. Due to the COVID-19 pandemic, we had to cancel the investigations.

The study was conducted according to the declaration of Helsinki and was approved by the local ethics committee of the University of Potsdam (Germany, approval no. 35/2018; 17 October 2018). All subjects were informed in detail and gave their written consent to participate.

### 2.2. Handheld Device to Record the Force Profiles during the MMT

The reaction force was measured by means of a novel wireless handheld dynamometer, which was created in a research and development project funded by the German Federal Ministry of Economics and Energy (BMWi; Project no. ZF4526901TS7). It was constructed and developed especially for clinical applications of MMTs to objectify the dynamics and kinematics during the test. The handheld device is based on a strain gauge unit (co. Sourcing map, model: a14071900ux0076, precision: 1.0 ± 0.1%, sensitivity: 0.3 mV/V, Hongkong, China), and kinematic sensor technology (Bosch BNO055, 9-axis absolute orientation sensor, sensitivity: ±1%, Stuttgart, Germany). All data were buffered with a sampling rate of 180 Hz, were AD converted and were sent via Bluetooth 5.0 to a tablet. A measuring software (based on National Instruments LabVIEW, Austin, TX, USA) saved the transmitted data. The sensor part was fixed between two interfaces, which were designed appropriately for the tester’s palm and the shape of the extremity of the tested subject to enable comfortable handling (Figure 1). The accuracy of the measurement was proofed in mechanical reliability measurements without subject (<1%).

### 2.3. Setting

During the MMT, the force profile between tester and subject is a result of their interaction. The setting of an MMT is to build up a closed kinematic chain, in which under all conditions the force is equal along all its links. In case of yielding in one of the chain connections, the force will generally be reduced in all parts of the chain. This means that even the strongest tester could not develop his or her maximum strength without an appropriate resistance. Since this study is focused on the reproducibility of the testers’ force profiles, it is not useful to execute the trials in a normal MMT setting. Subjects differ in their ability to resist. However, even if all measurements would be done with the same stable subject, one has to consider that there is a natural biological variability and, regarding the high number of trials, fatiguing effects are inevitable. For these reasons, a setting with a standardized maximal resistance was chosen by fixing the tested extremity (Figure 1). The MMT setting of the hip flexor muscles was used in this study, whereby the foot of the tested subject was placed against a wall to ensure a stable resistant but kept the properties of a tested real human extremity.

The tested assistant (male, age: 32 yrs., height: 185 cm, mass: 83 kg) was positioned supine on a foldable treatment table (height: 69.5 cm). The sole of the tested leg was located at a wall in a defined way with hip and knee angles of 90° and the anterior edge of the tibia in a horizontal position. The tested leg provided the typical elastic feeling of thigh muscles to the tester, but it was completely stabilized due to the contact to the solid wall. The examiner stood beside the table in a position according to the MMT procedure of the hip flexors and adjusted his or her stance to have a comfortable but also stable position during the MMT. The tester held the handheld device in his or her palm and contacted the assistant immediately proximal to the knee without giving pressure towards the condyles of the femur or the patella of the assistant. The contact position as well as the positioning of the subject’s foot was marked by tape strips to reproduce them properly. The pressure was then applied orthogonally from ventral against the distal end of the thigh via the device with the intention to extend the hip. As mentioned above, the hip extension was prevented by the fixation at the wall.

### 2.4. Procedure

After adjustment and positioning in the test setting, the instruction concerning the force profile was given to the testers. The experienced and little experienced testers were asked to perform the test as usual (intuitive MMT) using the handheld device. The beginners were introduced according to a first instruction of the MMT performance in the training course (see below). This included to rise the applied force for approximately three seconds merging into a plateau which should be held for one second. All testers were asked to develop a force high enough to make sure the muscle group would be classified as “stable”. A measurement leader gave all instructions, documented the process and possible appearing peculiarities and controlled the measurements including storage of the data.

Each tester performed 2–3 pre-trials to become familiarized with the setting. Subsequently, 10 intuitive MMTs were recorded with resting periods of approximately 15 s between them. No feedback about the progression of the force curves was given, neither verbally nor visually. After all the trials, a short interview was done with the tester about its feeling of the positioning, time course and forces.

### 2.5. Defined Force Profile for the MMT Measurements

To gather comparable items, it was necessary to choose one particular manner of testing. Therefore, a specific temporal force profile was defined. This was especially required, because the included beginners did not have any idea how to perform MMT. The beginners were instructed according to the initial instruction of the MMT provided in the basic course of the training program “Applied Kinesiology based Integrative Medicine” at the Academy of Health and Exercise Therapy prior to the measurements. All LitExp and Exp testers had already attended the training program.

The force profile is defined as follows (Figure 2): Phase 1 is to get in contact with the subject’s thigh and adjust a consistent but very low level of reaction force for some seconds. This is to create a starting level, as a first contact between tester and tested person. Phase 2 comprises an exponential increase of force in order to merge into a linear rise (Phase 3). At a force level, which is individually determined by the tester, the curve starts to flatten and passes over to a short plateau (Phase 4). During this period, the maximum force of the trial occurs. The testers were instructed to reach the maximum force within about 3–4 s, estimated by their own feeling during the test. The magnitude of the applied force was estimated by the tester with the intention to proof the hip flexors of the assisting person as “stable” during a usual (non-fixed) performed MMT.

### 2.6. Data Processing

Because no movement occurred in the performed setting due to the fixed leg, only the signals of the strain gauge unit were processed here. A linear spline interpolation was applied to ensure equidistant time channels (1000 Hz). The data were filtered using low pass Butterworth filters. Different filter degrees and cut off frequencies were used to address the different cutting approaches and parameters: (1) Cut-off frequency 3 Hz and filter degree 10 for cutting the curves from 0.5 s before force rise (using the first derivative of the force curve) to the maximum force value. (2) Cut-off frequency 16 Hz and filter degree 16 for cutting the curves from 4 s before 50% of the averaged maximum force of the ten trials (M_F_max_) to the end of trial. Thereby, the force profiles were lapped at 4 s at the same force amount to ensure a suitable cut of the force profiles.

In order to answer the abovementioned research questions concerning (1) the reproducibility of force profiles and (2) the differences between the groups Exp, LitExp and Beg, the following items were evaluated.

#### 2.6.1. Force Parameters

The starting force (F_start_ (N)) gives an impression about the initially applied force amount, which should be on a low level (see description of force profile). The arithmetic mean (M) and standard deviation (SD) of the first 500 data points (0–0.5 s interval) were calculated for each trial (F_start__M1 to F_start__M10). Furthermore, the M, SD and coefficient of variation (CV) of the ten F_start_ values (M_F_start_; SD_F_start_; CV_F_start_) were calculated. Cutting approach (1) was used.

The maximum force (F_max_ (N)) is defined as the highest value of each force curve (F_max__M1 to F_max__M10). The M, SD and CV of the 10 trials (M_F_max_; SD_F_max_; CV_F_max_) were calculated. Cutting approach (2) was used. Furthermore, the ratio of F_start_ to F_max_ was calculated. Therefore, the value of F_start_ was divided by the value of F_max_ for each curve and was averaged over the ten trials.

#### 2.6.2. Intraclass Correlation Coefficient

A possible parameter of estimating the accordance of force profiles is the intraclass correlation coefficient (ICC). The ICC(3,1) with absolute agreement was used in IBM SPSS 27 to evaluate the correlation between the 10 force profiles of each tester. Cutting approach (2) was used. Thereof, the data of each of the ten force signals was used from 2 s to the maximum value of the mean curve (F_max__Mean). The mean force curve is the average of the 10 force signals. Since each force profile has an increasing pattern, we expect very high ICC values overall in the groups.

#### 2.6.3. Normalized Mean Euclidean Distance

The aim is to receive a parameter of the deviation of the ten force profiles of each tester. The ICC or other correlation parameters might not be appropriate here, since all curves have a force increase and, therefore, the correlation will be high—also for low reliable force profiles. Therefore, the Euclidean distance was determined, additionally, to evaluate the distances between the 10 force profiles. Cutting approach (2) was used. For calculating the Euclidean distance, only the data points from 2.0 to 3.5 s were used to get an item to assess the deviation between the 10 trials within the exponential and first linear phase of the force profiles. The Euclidean distance (ED) (1) was calculated between the ten curves at each data point in this interval. Therefore, at each data point, the squared differences of each curve to another (M1 to M2, M1 to M3, …, M9 to M10) were summed up and thereof the square root was calculated for each data point q:(1)$${\mathrm{ED}}_{q}\;=\;\sqrt{\sum_{i=1,\;j=2}^{10}{\left(x_{q\_Mi}-x_{q\_Mj}\right)}^{2}}$$

i≠j, *q* = 2000 to 3500 data point (refers to 2.0 to 3.5 s). These ED_q_ (N) were averaged over all data points and relativized to the M_F_max_ (N), so that one normalized averaged ED (MED (%)) results per tester as a measure of distances between the 10 force profiles.

#### 2.6.4. Slope

The slope style is presumably one decisive parameter during the MMT. Therefore, the slope between the groups Exp, LitExp and Beg were considered as well as the reproducibility between the 10 trials of each participant. Different approaches were used:Slope of 2 s to Fmax (Slope_2max)

Using the cutting approach (2), the slope was calculated by m_2max_ = $\frac{y_{2}-y_{1}}{x_{2}-x_{1}}$, whereby the first data point (P_2s_) refers to the time (x_1_) and force (y_1_) values at 2 s and the second data point (P_max_) refers to the time (x_2_) and force (y_2_) values of the maximum of the corresponding force curve. The Slope_2max (N/s) was calculated for each curve M1 to M10 for individual reproducibility comparison and—for group comparisons the M, SD and CV of M1 to M10—was calculated per participant.

2.Slope XYZ in the linear section (Slope_XYZ)

A quasi linear part should arise after a smooth force increase in the course of the MMT. For evaluating this linear section, three points were defined: X = 40% of F_max_ of each curve, Y = 50% of F_max_, and Z = 60% of F_max_. The slopes between the points XY, YZ and XZ (N/s), respectively, were calculated by m*_XY_* = $\frac{y_{2}-y_{1}}{x_{2}-x_{1}}$ (YZ, XZ analogues) for each trial M1 to M10 as well as averaged over all ten curves. Cutting approach (2) was used.

3.Difference of slope YZ and XZ (Diff_YZ-XY)

This parameter was chosen to get an impression of the linearity in this section. In an ideal linear part, it should arise Diff_YZ-XY = m_yz_ − m_xy_ = 0. In case the second part increases steeper, the difference would be negative, in case the first part shows a faster increase, a positive difference would occur. Cutting approach (2) was used.

4.First derivative of the force curve

The first derivative represents the slope profile of the force curve, calculated by the difference quotient of each two consecutive data points. The M, SD and maximum (Max) of the first derivative from 0.5 s before force rise to F_max_ of each force curve was calculated for further comparisons. Furthermore, the M and SD over the 10 trials were calculated. Cutting approach (1) was used.

### 2.7. Statistical Analysis

All statistical procedures were performed with IBM SPSS 27 (Armonk, New York, NY, USA). Firstly, all data were checked for normal distribution by means of the Shapiro Wilk test. Secondly, the following considerations were performed:

The reproducibility of M1 to M10 was checked by the analysis of variance for repeated measurements (RM ANOVA) (for parametric data, normal distribution fulfilled) within the groups 1 (Exp), 2 (LitExp) and 3 (Beg). In case the normal distribution was not fulfilled in more than two data sets (since there is evidence that the RM ANOVA is robust [64]) or the Levène test of variance homogeneity was not fulfilled, the Friedman test was performed. In case the Mauchly test of sphericity was significant, the Greenhouse Geisser correction was applied. As a post-hoc test, the Dunn–Bonferroni correction was performed (p_adj_). The group differences between Exp, LitExp and Beg were verified by a multivariate analysis of variance (MANOVA) (for parametric data) in case the variance homogeneity (Levène test) was fulfilled. If the data were not normally distributed or the variance homogeneity was not fulfilled the Kruskal–Wallis test was performed. The significance level was always set at α = 0.05. In case of significance, the effect size was calculated by either eta-square (η^2^) for the RM ANOVA or the MANOVA, respectively, or by Pearson’s r = $\left|\frac{z}{\sqrt{n}}\right|$ for the values of Dunn–Bonferroni post-hoc test of the Friedman or for the Kruskal–Wallis test, respectively.

## 3. Results

The data of each participant, trial and parameter can be found in the Appendix A. The ten force curves of each participant sorted by the groups Exp, LitExp and Beg are shown in Figure 3, Figure 4 and Figure 5. As can be seen, especially the experienced testers have a reproducible force profile (with two exceptions), but also in the groups of LitExp and Beg at least one or two testers show a good reproducibility based on descriptive considerations. Especially the group of Beg shows some testers with inconsistent force profiles (Figure 5). Furthermore, it is visible that the force profiles differ with respect to the above described phases. The first six of nine experienced testers show the profile as described above. The others deviate, especially with respect to the slope style. Furthermore, the Fmax seems to differ between the testers irrespective of the groups.

### 3.1. Intraindividual Comparisons of the 10 Repeated Trials

#### 3.1.1. Force Parameters

The starting force (F_start_) shows no significant difference in the RM ANOVA or the Friedman test between the 10 trials within the groups Exp, LitExp and Beg (*p* > 0.05). The maximum force F_max_ differs significantly between the 10 trials for the Beg (*F*(3.6,39.6) = 2.881, *p* = 0.039, η^2^ = 0.208) and the LitExp (*χ^2^* = 18.218, *p* = 0.033), not for the Exp (*F*(3.7,29.5) = 1.177, *p* = 0.322). Figure 6 displays the individual M, SD and 95%-CI of the F_max_ of the 10 trials of each tester. The CV of F_max_ between M1 to M10 ranges for Beg from 3.3 to 13.2% (M ± SD: 7.7 ± 3.0%), for LitExp from 3.7 to 16.0% (8.5 ± 4.3%) and for Exp from 4.6 to 12.1% (7.6 ± 2.5%). The ratio of F_start_ to F_max_ does not differ significantly between the 10 repeated trials within the groups Exp, LitExp or Beg (*p* > 0.05).

#### 3.1.2. Slope

Figure 7 illustrates the intrapersonal M, SD and 95%-CI of the Slope_2max of the 10 trials of each participant, which shows no significant result for the RM ANOVA between M1 to M10 within the groups Exp, LitExp and Beg (Exp: p_adj_ = 0.238; LitExp: p_adj_ = 0.180; Beg: p_adj_ = 0.276). For all groups, the slope in the linear section XYZ shows no significant difference in the Friedman test (Beg) or the RM ANOVA (Exp, LitExp) between the 10 trials (Table 2). The parameters of the first derivative do also not differ significantly for the 10 repetitions within each group (*p* > 0.05). The data of each tester, trial and parameter can be found in the Appendix A.

### 3.2. Group Comparisons between Experienced, Little Experienced and Beginners

Table 3 displays the results of the inter-group comparisons between Exp, LitExp and Beg (Kruskal–Wallis test) concerning the force parameters, the ICC(3,1) and the MED. The results of group comparisons (MANOVA) concerning the slope parameters are given later (Table 6). The values of each tester, each trial and each parameter can be found in the Appendix A.

#### 3.2.1. Force Parameters

The Beg group shows a significantly higher starting force level (F_start_) compared to the Exp group (p_adj_ = 0.029, *r* = 0.56). The LitExp group is sorted between both other groups, but shows no significance compared to them (Table 3, Figure 8a). The maximum force (F_max_) does not differ significantly between the three groups, although the Exp show the highest F_max_ with averagely 227.17 ± 70.26 N vs. LitExp with 153.13 ± 89.09 N vs. Beg with 175.26 ± 69.53 N (*p* > 0.05) (Table 3, Figure 8b). The ratio of F_start_ to F_max_ is significant in the Kruskal–Wallis test between Exp and Beg (p_adj_ = 0.005, r = 0.68) and not significant between Exp and LitExp (p_adj_ = 0.150) and between Beg and LitExp (p_adj_ = 1.000) (Table 3, Figure 8c).

#### 3.2.2. Intraclass Correlation Coefficient (ICC(3,1))

The Exp group shows the highest value of ICC(3,1) with averagely 0.984 ± 0.02, LitExp follows with 0.976 ± 0.012 and Beg have the lowest ICC values with averagely 0.956 ± 0.046 (Figure 9a). Six Exp show values of ICC(3,1) ≥ 0.989, the others show lower ICCs. The Exp tester with lowest ICC(3,1) amount to 0.9310 (Exp_9). Five LitExp (*n* = 5) are ranged between 0.979 and 0.988. The lowest ICC(3,1) = 0.834 was reached by one Beg (Beg_1). The Kruskal–Wallis test between Exp, LitExp and Beg turned out to be still significant (*p* = 0.040). The pairwise comparisons resulted in a significant difference between Exp and Beg (*p* = 0.018), which turned out to be not significant anymore regarding the adjusted *p*-value of the Dunn–Bonferroni post-hoc test (p_adj_ = 0.54, r = 0.52). The 95% confidence intervals (Figure 9a) illustrate the comparable high spreading of Beg. Referring to the results, the testers are sorted into three groups concerning the amount of ICC(3,1): (1) high ICC(3,1) ≥ 0.989, (2) moderate ICC(3,1) ranges from 0.979 to 0.988 and (3)—for the needs of the present investigation—low ICC(3,1) < 0.979 (Table 4). In order to determine a border of ICC(3,1) for a considerably sufficient high reliability, ICC(3,1) = 0.989 is suggested. Six experienced testers (Exp_1, 2, 3, 5, 6, 8) and two beginners (Beg_4, 11) would be denominated, thereby, as highly reliable. An ICC(3,1) from 0.979 to <0.989 would still be regarded as sufficient reliability for a beginner. An experienced tester should show higher amounts of ICC. ICC values of <0.979 would be assessed as not sufficiently reliable for the repetitions of force profiles using the MMTs, also not for beginners. One experienced tester (Exp_9), three little experienced (LitExp_1, 5, 7) and six beginners (Beg_ 1, 2, 3, 7, 9, 10) would fall into this group. We are aware that the suggested border of ICC value is still very high for common investigations [65]. However, the consideration of increasing force profiles using the ICC needs the definition of new thresholds due to the characteristics of data. Therefore, the abovementioned three areas of ICC are initially suggested for the evaluation of reproducibility of force profiles. By comparing this categorization with the force profiles seen in Figure 3, Figure 4 and Figure 5, a close match is visible, which confirms the lower reliability especially of Exp_9, but also of Exp_7 and 4 (Figure 3) and the considerably high reliability of Beg_4 and Beg_11 (Figure 5).

#### 3.2.3. Normalized Mean Euclidean Distance (MED)

The Exp group shows averagely the lowest MED with 26.5 ± 9.1% (range: 18.2–44.9%), the LitExp has significantly higher values of 40.72 ± 8.3% (range: 29.2–57.1%) and the highest MED values emerge for the Beg with averagely 47.51 ± 19.92% (range: 22.8–78.8%). The Kruskal–Wallis test shows significant differences between Exp and LitExp (p_adj_ = 0.047, r = 0.585) and Beg (p_adj_ = 0.015, r = 0.612), respectively. The LitExp and Beg do not differ significantly (p_adj_ = 1.0) (Table 3, Figure 9b).

We also want to suggest thresholds for the MED resulting from the reached values of the testers: (1) high reproducibility: MED < 26%; (2) moderate reproducibility: MED = 26–40%; low reproducibility: MED > 40%. For interpretation of the values, it has to be noted that the MED is not comparable to a commonly calculated deviation of values as the coefficient of variation or the like. For determining the MED, the root of the sum of 45 squared distances (each of the *n* = 10 force profiles to the other) was calculated for each data point and then was averaged over all data points. That is why the values are considerably higher compared to other parameters of variability and need to be interpreted differently. The thresholds defined above, result in a similar sorting (Table 5) compared to the ICC(3,1) (Table 4): the same *n* = 6 experienced testers (Exp_1, 2, 3, 5, 6, 8) and *n* = 2 beginners (Beg_4, 11) fulfill the recommendations of a low MED, thus, show a high reproducibility. The highest MED of experienced tester amounts to 45% (Exp_9). The highest value of all testers is MED = 78.8% (Beg_3).

#### 3.2.4. Slope Parameters

The overall slope from 2s to the maximum force (slope_2max) amounts 72.03 ± 27.16 N/s for Exp (range: 22.15–121.70 N/s), 37.92 ± 18.62 N/s for LitExp (range: 14.43–76.46 N/s) and 45.06 ± 21.27 N/s for Beg (range: 17.61 – 80.18 N/s) (Figure 10a, Table 6). The ANOVA shows significant differences between the groups (F(2,26) = 5.635, *p* = 0.009, η^2^ = 0.302). In Bonferroni post-hoc test the Exp and LitExp (p_adj_ = 0.014) as well as Beg (p_adj_ = 0.036) differ significantly, whereas LitExp and Beg do not (p_adj_ = 1.0).

The second parameter for the overall slope is the M, SD and Max of the 1st derivative of each curve, which differs significantly for the M (*p* = 0.005, η^2^ = 0.334), the SD (*p* = 0.010, η^2^ = 0.296) and the Max of 1st derivative (*p* = 0.020, η^2^ = 0.260) between the groups. The Exp show the highest values of M, SD and Max (Table 6, Figure 10). The post-hoc Bonferroni test reveals significant differences for Exp vs. LitExp (p_adj_ = 0.008 to 0.024) and Exp vs. Beg (p_adj_ = 0.022 to 0.042 for M and SD of 1st derivative; Max n.s.). The LitExp and Beg show no significant difference (Table 6, Figure 10b).

The M, SD and CV of the slope in the linear section XYZ and the results of group comparisons are given in Table 6. The slope_XYZ shows significant differences between the groups analyzed by MANOVA (XY: *p* = 0.016, η^2^ = 0.28; YZ: *p* = 0.007, η^2^ = 0.32; XZ: *p* = 0.007, η^2^ = 0.32). Thereby, the post-hoc test reveals that the Exp group has significantly higher slope values in all sections compared to both other groups (p_adj_ = 0.004 to 0.013), whereby the LitExp and Beg show no significant difference (*p* > 0.05) (Table 6, Figure 11). The difference between the first and second linear section (Diff_YZ-XY) is significant for the comparison of Exp and Beg (*p* = 0.020; Figure 11d), but not for the adjusted *p*-value (*p* = 0.059; Table 6). It is lowest for the Exp group. This indicates that the Exp has the higher slope in the second part of the linear section YZ, whereby the LitExp shows the best linearity in this section and the Beg perform a higher slope in the first section XY compared to the second section YZ.

## 4. Discussion

In the present study, the aim was to investigate the force profiles of manual muscle testing with regard to (1) the reliability of the testers performing 10 repeated trials and (2) the force profiles concerning the test experience. The main difference to the “real” manual muscle test was the fixed tested extremity, so that the influence of the patient’s capacity of Adaptive Force was eliminated. Therefore, only the force profile of the tester was the objective here.

### 4.1. Reproducibility of Force Profiles within the Groups of Beg, LitExp and Exp

The reproducibility is one of the main requirements to use the MMT in practice to get valuable diagnostic results. In this study, it is visible in general that the testers—irrespective of the test experience—show different force profiles with respect to the starting force, maximum force and the slope. However, the question of the intra-tester reproducibility is essential, too. Looking at the maximum force of the testers force profiles over 10 repeated trials, the results showed that only the experienced testers are able to statistically reproduce the maximum force in 10 repetitions, whereas the little experienced and beginners showed significant differences (*p* < 0.05). Nevertheless, in all groups there are examiners with higher and lower reproducibility, respectively. We suggest accepting a variation of the F_max_ < 8% (for justification see below); 19 of the 29 testers would fulfill these requirements (5 Exp, 6 LitExp, 8 Beg).

As a parameter for the reproducibility of force curves, we used the normalized mean Euclidean distances (MED) and the ICC(3,1) between the 10 curves. The results show that also experienced testers partly might have a rather limited reproducibility (Exp_4, 7, 9), in showing a MED of > 26% (30%, 36%, and 45%, respectively) and an ICC(3,1) < 0.98 (0.986, 0.979 and 0.931, respectively). Looking at the Beg the MED and ICC(3,1) would designate the same two beginners to the testers with good reproducibility (Beg_4, 11) with MED = 23% and 25%, respectively, and ICC(3,1) = 0.993 and 0.992, respectively. As mentioned in the result section, an ICC(3,1) ≥ 0.989 is regarded here as a high reproducibility. Therefore, the force profiles of no little experienced tester would be regarded as highly reproducible. However, one little experienced tester (LitExp_2) shows a MED of 29% with an ICC(3,1) of 0.983. The considerably good MED is probably due to the very high maximum force value of 338 N of this tester, since the MED includes the normalization to F_max_. The ICC(3,1), which is not normalized to the F_max_, is considerably lower for this tester. Two other LitExp testers (LitExp_4, 6) also show an ICC(3,1) of 0.983 but have higher MED values (38% and 37%).

The combination of ICC(3,1) and MED might be suitable quality criteria for reproducibility with the strict borders of MED ≤ 26% and ICC(3,1) ≥ 0.989. Most of the experienced testers would meet these quality requirements (Exp_1, 2, 3, 5, 6, 8) as well as two beginners (Beg_4, 11). As a minimum requirement for the repetition of 10 trials for testers, we suggest a border of ICC(3,1) of 0.979 and a MED of 40%. Eight testers would fulfill both minimal specifications (Exp_4, 7; LitExp_2, 4, 6; Beg_5, 6, 8) and a further eight testers would fail to pass them (Exp_9, LitExp_5, 7; Beg_1, 2, 3, 9, 10). However, there are some testers, who meet only one of both criteria, ICC or MED, respectively (LitExp_1; Beg_7 would fulfill MED but not ICC; LitExp_3, 8; Beg_12 vice versa). These results point out that both parameters cover different aspects of the reproducibility but work complementing each other. The allocation of the involved examiners regarding the criteria strongly indicates that the combination of ICC and MED could allow a nuanced distinction of different skills to generate reproducible force profiles.

Summarizing those results, the most experienced testers are able to reproduce their force profiles appropriately, but not all of them. Surprisingly, even some beginners seem to be able to generate reproducible force courses over time with satisfying quality. However, the majority of the beginners or little experienced testers are not able to generate highly reproducible force profiles. That underlines the necessity of practicing the MMT to acquire and retain a proper motor performance during the MMT. Despite a lot of practicing time, there are also some experienced testers with poorer reproducibility. Besides experience, the personal sensorimotor skills of the testers obviously seem to be essential. This underlines the need to assess the reproducibility of the force profiles. An assessment should be done as early as possible in the course of learning by means of objective measuring.

Executing the real MMT with a patient, the motor performance gets more complex for the tester, since there is an interaction between tester and patient. According to Doyon and Ungerleider [66,67], motor skill learning has to be differentiated into motor sequence learning and motor adaptation. The latter includes the “capacity to compensate for environmental changes” [66]. This surely is relevant for the performance of MMT in interaction with a patient. However, the results of the present study show that there is the need to practice the force profile during the MMT firstly without a patient to learn and optimize the force profile, thus a motor sequence learning of the acquired force profile. A person will never be able to test like a machine; however, the intention and aim of each practitioner should be to perform the MMT as reproducibly as possible by learning this complex motor skill until it gets fully automatic.

### 4.2. Group Differences regarding the Course of Applied Force with Respect to the Test Experience

The starting force showed higher values for the beginners than for the experienced or little experienced testers. Therefore, it is assumed that the beginners could be less sensitive in the first contact phase, where a soft contact to the patient has to be built up prior to the force rise. The maximum force did not differ significantly between the groups. However, the experienced testers tend to applicate a higher maximum force, whereas testers with no or little experience show a mean maximum force on a similar lower level. This indicates that experienced testers seem to be more cautious and sensitive at the beginning but robust at the end of the MMT. The ratio of starting force to maximum force show differences between experienced testers and beginners (*p* = 0.005). The ratio of little experienced testers is located between beginners and experienced testers. That shows that the force application could be a parameter of interest during the learning process. Provided that the experienced testers optimize their testing procedure in the course of successful applications, beginners could be instructed to start sensitively but encouraged to end with a more powerful finish.

It has to be taken into account here, that the gender distribution is not similar in the groups Exp (m = 7, f = 2), LitExp (m = 3, f = 5) and Beg (m = 3, f = 9). This could especially have an influence on the maximum force. The maximum force of females over all groups amounted averagely 152.14 ± 49.64 N and for males 226.02 ± 90.09 N and differed significantly in the t-test for independent samples (*p* = 0.016). However, looking at the single groups, the force levels of male and females in the Exp group are similar with 226.07 ± 75.73 N and 231.00 ± 71.06 N, even slightly higher for females. Due to the high variance of males, also the group of LitExp shows no significant difference between females (129.07 ± 22.27 N) and males (193.22 ± 151.43 N, *p* > 0.05). Only the Beg show a significant difference concerning the force levels between males (258.72 ± 73.11 N) and females (147.44 ± 42.75 N) (*p* = 0.008). Those results indicate that the maximum force also is a parameter of the MMT, which depends on the test experience. With a long test experience, the gender seems not to be a relevant factor for the maximum force anymore. However, it must be pointed out that those results are based on rather small sample sizes. Since the MMT is not a test of maximum strength, this result is not surprising. All testers were told to use as much force as they would apply to test a young athletic male. Testers with no or little experience might possibly not have the feeling of how much force is necessary to test a young athletic male participant.

All slope parameters showed significantly higher values for experienced testers compared to both the other groups. For little experienced testers and beginners, the slope is comparable low. Especially in the third phase, the linear section, the difference of slope in the part YZ to part XY is lowest in Exp with a negative arithmetic mean value. Whereby there is no difference between the LitExp and Beg groups, indicating that the force course in the sense of slope progression is developed with several years of test experience. This suggests that the little experienced testers and beginners seem to be more cautious in applying the force rise, whereas experienced ones obviously challenge the subject harder in the upper half of force development. This reasoning is underpinned by the more rapid rise of and the higher maximum force applied by experienced persons. This, of course, might also be a random effect due to the small sample sizes in the groups. Therefore, the sample sizes have to be increased to get a better impression of the patterns of force profiles.

Finally, the following major results can be summarized:Experienced testers are able to generate more reproducible force profiles compared to little experienced testers and beginners.Experienced testers start with a more sensitive contact but followed by a steeper rise of force up to a higher maximum force.There is a large heterogeneity between the force profiles of different testers regarding the starting force, the maximum force, the slope, and the reproducibility—this is lessened amongst experienced testers, but nevertheless present.Independent from the influence of experience, the personal sensorimotor skills seem to be a major factor regarding the ability to create a proper force profile.

It is assumed that the force profiles of different experienced testers will vary significantly also regarding a larger sample size, since each tester develops its own style in the course of practicing. This is a point which leads to the understandable criticism of the MMT. Different force styles could result in different judgements in testing the same patient. Reaching for more acceptance in medicine and science, neurophysiological considerations on the essence of the MMT might support (see Section 4.3). Based on those reflections we will suggest a standardization of force profile for assessment in training courses (see Section 4.4).

### 4.3. Neurophysiological Considerations regarding the Force Profile

The shape of the force profile must be adjusted according to the aim of what should be tested by the MMT. However, there is a lack of clarity in literature and some ambiguity amongst different authors concerning the notion of what kind of motor function is tested. It was suggested that the MMT is a method of investigating spinal dysfunction, but also the central and the peripheral nervous system [62]. As mentioned in the introduction—from our point of view—the MMT examines the capability of the neuromuscular system to adapt to external varying forces, thus, the term Adaptive Force seems to be appropriate and was introduced [30,31]. Thereby, central and peripheral structures of the neuromuscular system have to cooperate closely to secure the appropriate adaptation to the external circumstances.

In the setting of the MMT, tester and patient interact, optimally only in a quasi-isometric equilibrium, despite the tester’s varying force application. The tester performs a pushing isometric muscle action (PIMA) and the patient performs a holding isometric muscle action (HIMA), whereby he or she must just react and adapt to the applied force of the tester. If the patient can resist properly, the interaction stays isometric until the end of the test. If the patient is not able to adapt adequately, the patient’s muscle will lengthen during the force rise applied by the tester; thus, it will merge into eccentric muscle action. If a muscle gives in before its maximum force capacity is reached, the subject’s sensorimotor system has failed to adapt. This could be due to the time problem of the neuromuscular system during adaptation to external forces, as mentioned in the introduction. The patient is asked to stabilize a given position of a limb, while the rising external force tends to move the limb out of its position. The signals of an initial deviation from the proprioceptive and surface sensory system cause long latency responses with a runtime of about 100 ms [58,59]. If an external input is varying, the relevant motor corrections follow also with latencies of 80–110 ms [32,57]. The corresponding muscle must increase its force to compensate the deviations and to maintain the position and muscle length. However, when the correcting response arrives after about 100 ms, the external force already has increased further. Therefore, during a continuous external force increase an ongoing forward control mechanism is necessary in order to implement adjustments including the anticipatory rise of force after 100 ms. The task of forward controlling is presumably undertaken by the olivocerebellar circuitry [42,48,50,51,60,68]. In this process the ability of the cerebellum to predict connected motoric events in cooperation with the inferior olivary nucleus (ION) is utilized. The ION provides the rhythmic neuronal time signal to enable temporal coordinated movements [42,48,60]. We suggest that the Adaptive Force, which is tested by the MMT, is based on a forward controlling of the muscle tension and length in the rhythm of the central control circuits of 10 Hz, which is triggered by mismatches of the positioning sensors. To avoid running behind the permanently changing force requirements, the neuromuscular system must anticipate and adjust to the expected force rise. Otherwise, no isometric equilibrium between tester and patient under changing applied forces would be possible. During the MMT (“break test” mode) with a patient, the aim is not to test the maximal strength of the patient or to “demonstrate” that the patient is not able to resist the external force. The aim is to assess if the adaptation capacity of the neuromuscular system functions in a normal way.

In the setting of a real MMT including the tester and a subject with free, non-blocked limbs, a phenomenon arises, which underpins the theory of the 10 Hz-driven adjustment circuitry discussed above. In case the subject can adapt adequately to the tester’s force rise, a mutual oscillation appears. This oscillation can be found in dynamometric and kinematic measures and shows a stochastically distributed frequency of around 10 Hz. Obviously, both involved persons get together in a synchronized way and the coupling runs with 10 Hz. These coherent oscillations of around 10 Hz between two isometrically interacting persons were also found independent of the MMT [69,70]. In contrast, if the subject yields during the MMT force rise, those oscillations do not arise or appear only sparsely. Figure 12 shows the exemplary curves of force (above) and gyrometer signals (below) during three MMT measurements using the handheld device (a still unpublished pilot study). An experienced tester performed the MMT of the hip flexors of one healthy female participant (44 years, height: 173 cm, body mass: 77 kg) under different conditions: “stable” (blue and green) vs. “unstable” (red) caused by different odors (neutral: blue; pleasant: green; disgusting: red). The oscillations of force (above) and gyrometer signals (below) can be seen in the green and blue curves under stable conditions from 4 s to the maximum force (above). In contrast, the red lines show only poor oscillations (especially in the force signal) during this period.

Furthermore, the gyrometer signal during the unstable status (Figure 12 below, red) clearly shows the start of the yielding of the participant’s limb at an Adaptive Force of 96 N. This is referred to as the maximum isometric Adaptive Force (AFiso_max_) force. However, in this case the force continues to increase during and despite of muscle lengthening up to a maximum eccentric Adaptive Force value of AFmax_ecc_ = 158 N. As can be seen, under stable conditions with neutral or pleasant odors (blue and green lines), the AFmax is reached within a maintained isometric position and, therefore, the AFmax = AFiso_max_ with values of 146 and 137 N, respectively. The gyrometer signals (Figure 12 below), thereby, clearly show the quasi-isometric status of the limb until the AFiso_max_ is reached. Summarizing, under unstable conditions the AFmax indeed is the highest, but it is reached during muscle lengthening. The AFiso_max_ shows a considerably lower value and amounts—in this particular case under unstable conditions—to only 65% and 70%, respectively, of the AFiso_max_ under stable conditions. At this point the adaptation seems to fail and, in turn, it is the range of intensity where the oscillations appear in a MMT under “stable” conditions. This behavior is observable regularly in other subjects. Thus, probably the 10 Hz driven adaptation may play a role for maintaining a stable holding isometric muscle action during adaptation to an external force rise.

Provided the MMT examines a forward controlling of the neuromuscular system by adapting to external increasing forces (AF), it still remains unclear how the shape of an adequate force profile should be characterized for testing this capacity. A first suggestion of a standardized force profile was already made in the method section and is based on methodological but also on neurophysiological considerations. The force-time line regarding the mechanisms of forward controlling seems to be of particular relevance in defining a tester’s force profile. In order to stabilize the position, the tested neuromuscular system must anticipate the expected force 100 ms later. This will not be very challenging during a constant force input. During a linear force rise the stepwise increase will be equal every 100 ms. Therefore, it might also be not too complicated for the neuromuscular system to adapt to a linear rise. Then again, under nonlinear requirements the anticipation will be more challenging because of the altering width of the consecutive steps. Particularly, rapid accelerations of the force rise are characterized by largely altering steps of increase. On this account, we consider such periods during the MMT as vulnerable phases. Therefore, it is suggested for the tester’s force rise to start smoothly by an exponential increase with small force differences and to merge into a linear phase with a higher, but rather similar delta force.

### 4.4. Proposal for a Standardized Force Profile

We propose the force profile shown in Figure 13 as a possible model, which is put up for discussion here. We are aware that each tester works with his or her own force profile and procedure of MMT. However, with the aim to standardize the MMT, a first step should be taken. Surely, some aspects will be discussed within and between the different groups of MMT practitioners, for instance the duration of the test. As Conable and Rosner (2011) stated, many testers perform MMTs within 1 s or less [20]. The authors suggest that testers might consider a duration of 3 s or longer, especially if the test reveals equivocal results [20].

The force profile displayed in Figure 13 was generated by averaging the total 20 curves of tester 1 and 2 of the present study (Figure 14). These profiles were chosen since tester 1 and 2 are experienced teachers in the training program. Furthermore, the force profiles of these testers show not only a high reproducibility within each tester, but also comparing the profiles between both testers (ICC(3,1) = 0.989; Cronbach’s alpha of 1.0). These high values of inter-tester reproducibility proof a good similarity between both individual styles and, therefore, the averaged temporal course of force of those testers are used for the proposed model. Eventually, the profiles were chosen, because they contain all parts of the suggested four-phase approach in a clear manner.

Naturally, the profile must be adjusted according to the physical conditions of different patients (here: male, 32 years, 185 cm, 83 kg). In all cases, the following four phases should appear. Phase 1: A coupling of 1–2 s, in which the patient and the tester get in first contact for a smooth equilibrium at a very low but stable force level (<20 N); it should not exceed 5% of the F_max_. Phase 2: The transition from the low force level to the linear rise (from phase 1 to 3) seems to be the most vulnerable period because it requires an exponential increase. According to the forward controlling under nonlinear conditions, the patient should get the chance to adapt to the increasing force rise by applying manageable delta forces. The amount of force differences for the here presented profile along the consecutive 0.1 s steps ranges from 0.9 N at the beginning up to 9 N at the end of the exponential phase (duration ~1.5 s). Phase 3: The quasi-linear force rise with an increase of about 10 N per 0.1 s (duration ~1.5 s). Phase 4: Transition to the maximum force and short plateau (duration ~0.7 s). In a healthy person, this plateau can be hold for 1–3 s on a high force level.

Those phases are proposed for the force rise of the tester; however, in interaction with a patient, the profile depends not only on the tester. This force profile can only be created when the patient is able to maintain a stable isometric resistance during the whole process. As mentioned above, usually in phase 3 oscillations of about 10 Hz arise (Figure 12), which are interpreted as a sign of proper interaction reflecting the rhythmic neuromuscular motor control by the central structures. In case of a failed adaptation, the patient will not be able to build up a force plateau on a considerably high force level. Usually, in some patients, already during phase 2, more often during phase 3, the patient merges into an eccentric muscle action, the muscle lengthens and the involved limb yields (as can be seen Figure 12).

This again underpins the concept of MMT as a measure to assess the ability of sensorimotor adaptation but not to detect the maximum force. It seems to be crucial to identify the breaking point, when—in case of poorly functioning adaptation—a muscle starts to yield. For this purpose, it is necessary that the external force application by the tester surpasses the force amount of a possibly occurring breaking point, but it does not urgently need to reach the maximum force of a patient. At this point the largely differing averaged F_max_ between the testers in the present study is of relevance. Nearly half of them stay below 150 N. In case there is a breaking point above this threshold it would not be detected by these testers. This would lead to false negative test results. This reasoning substantiates why a minimum force level should be reached by the tester. It should be adequate to the patient’s physical constitution; high enough to exceed a possible breaking point, but not necessarily reaching the maximum force of the patient’s muscle; but yet, of course, not exceeding the maximum strength of the patient. The latter would lead to false positive test results.

Still, this argument stays hypothetically because it is based on the theory of the Adaptive Force including the muscle yielding under specific circumstances at submaximal force levels. Although there is some published research on this topic [30,31], more evidence is needed. Additional publication is currently in preparation.

## 5. Conclusions

The results show a broad heterogeneity of force profiles between different testers regarding starting force, maximum force, slope and duration, which could lead to differing judgements. Therefore, professional societies and communities should figure out what a good profile in practice should look like. Suitable testing styles should be taught standardized in training courses. Initially, it should be practiced applying a standardized force profile against a stable resistance. As it is well known, motor learning (of the art to test) requires a number of correct—and therefore corrected—repetitions. This must be controlled by objective measurements, for instance using a handheld device.

Furthermore, the results reveal an influence of the experience on the intraindividual reproducibility of the personal force profile. Therefore, the risk of false judgements of the patient’s muscle function will be higher in beginners and in those less experienced, too. This detriment could be probably compensated by systematic practices supported by objective measurements. We suggest a standardized assessment of the reliability of the individual force profile based on a number of repeated tests (for instance *n* = 10) and using statistical parameters like the combination of ICC and MED as we used in the present study. An assessment like this could be part of the examination after training courses. The result could be mentioned, for example, in the certificate.

The introduction of objective measuring into the clinical practice of MMT would be a quality leap. It could lead to more confidence and acceptance regarding the judgements. For this purpose, it is inevitable to record kinematic data simultaneously with the force measurement. The quantification of the maximum isometric Adaptive Force (AFiso_max_)—especially in relation to the maximum Adaptive Force (AF_max_)—would generate metric data which can be used for diagnostics or follow-up. Moreover, the tester would get a feedback about the quality of his or her testing.

Based on the neurophysiological considerations, it is concluded that the MMT examines the neuronal control of motor function in a specific adaptive way. We support Walther’s definition, after which “most muscle tests (…) do not evaluate the power a muscle can produce; rather, they evaluate how the nervous system controls muscle function.” [19]. In contrast to his own definition, Walther used the terms “strong” and “weak”, respectively, to judge the MMT [19]. Because this wording could be misunderstood (and indeed is frequently in daily practice) we suggest the notions “stable” vs. “unstable”. Of course, other denominations like, for instance, “resisting” vs. “relenting” are also thinkable.

Eventually, the use of objective measuring in combination with the MMT could be an approach to investigate possible causations lying beyond the clinically observed muscle “weakness” relating to many health issues (as mentioned in the introduction). In case of absence of neurological or muscular pathologies, functional mechanisms like nociceptive reflexes or psychogenic inhibitions should be discussed and investigated.

## 6. Patents

The patent specification (German and PCT application) of the handheld device reported in this manuscript was disclosed in October 2020. (patent applicator: University of Potsdam, Germany; Reference no. DE 10 2019 119 862 A1; WO 2020/208264 A1).

## Figures and Tables

**Figure 1 diagnostics-10-00996-f001:**
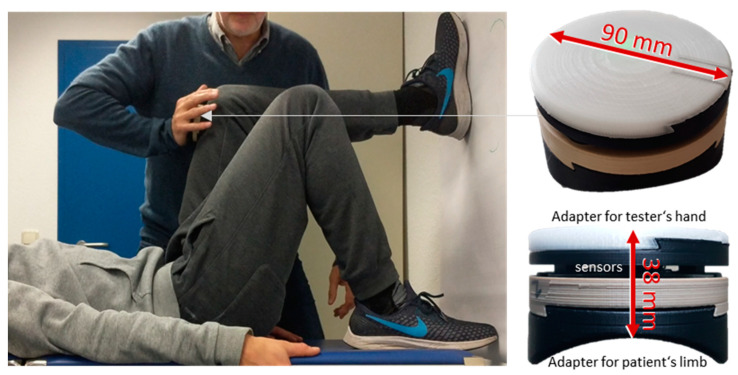
Setting of the manual muscle test (MMT) of the hip flexors with fixed leg including the handheld device. The assistant’s leg is fixed against the wall to ensure a stable position also after repeated trials. The tester’s forearm is in a rectangular position to the rectus femoris muscle. The handheld device is located between the tester’s hand and the assistant’s distal thigh to measure the force profile produced by the tester. On the right side, the handheld device with two adapters is illustrated: one for the palm of the tester (white) and one for the limb of the patient (black), which is shaped according to the contours of the limb. Between the adapters, the sensor technology is placed. Dimensions of the handheld device are given in mm.

**Figure 2 diagnostics-10-00996-f002:**
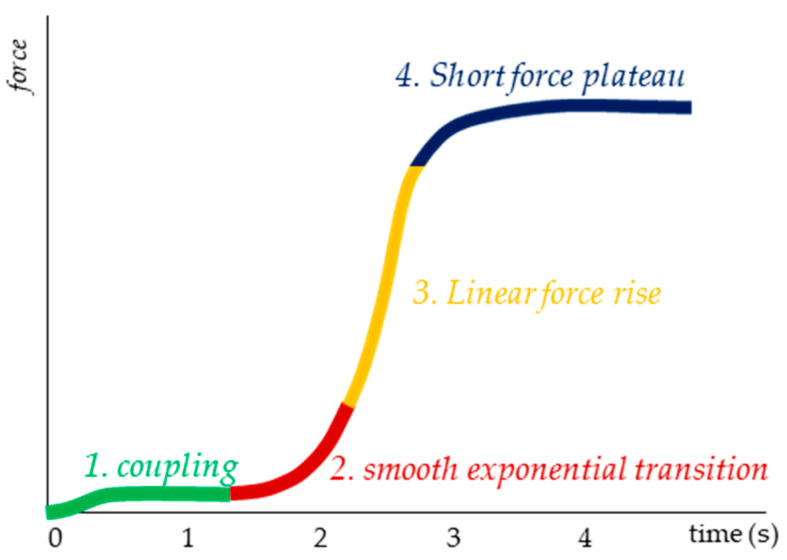
Schematical force profile including the phases of force rise.

**Figure 3 diagnostics-10-00996-f003:**
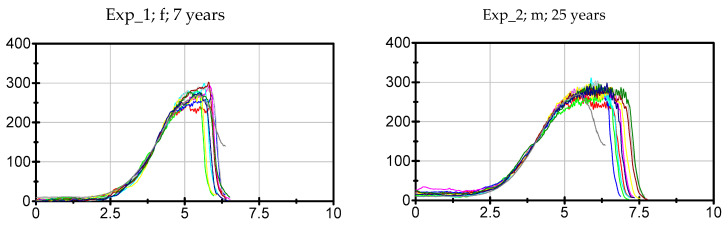
Force profiles of experienced testers. Displayed are the 10 individual force profiles of the *n* = 9 experienced testers (Exp). The grey curve displays the suggestion of a standardized force profile (see discussion). To have a better comparison, the *x*- and *y*-axis are standardized (0–10 s and 0–400 N). The ID (Exp_no.), gender (f = female, m = male) and the years of experience are given per tester.

**Figure 4 diagnostics-10-00996-f004:**
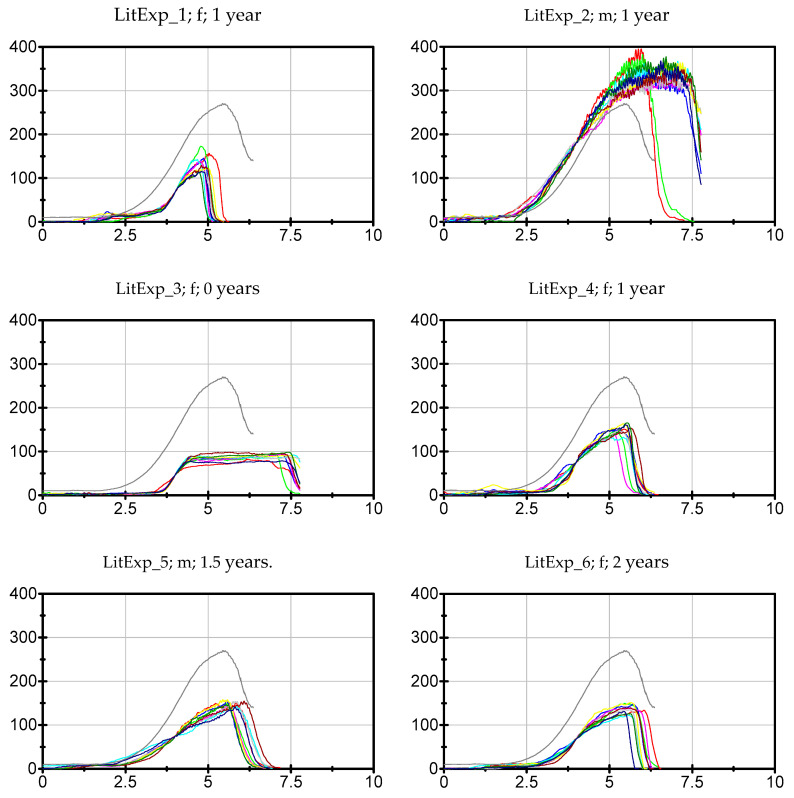
Force profiles of little experienced testers. Displayed are the 10 individual force profiles of the *n* = 8 little experienced testers (LitExp). The grey curve displays the suggestion of a standardized force profile (see discussion). To have a better comparison, the *x*- and *y*-axes are standardized (0–10 s and 0–400 N). The ID (LitExp_no.), gender (f = female, m = male) and the years of experience are given per tester. The one tester with zero years of experience had successfully participated at the basis course of “Applied-Kinesiology based Integrative Medicine” in 2019 (3 month prior to measurement) and since then practices with colleagues.

**Figure 5 diagnostics-10-00996-f005:**
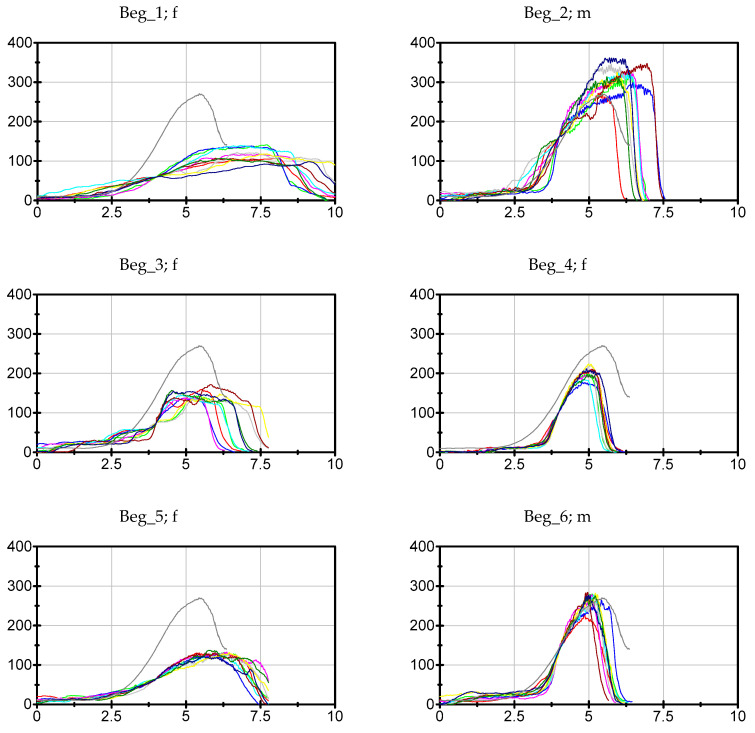
Force profiles of Beginners. Displayed are the 10 individual force profiles of the *n* = 12 beginners (Beg). The grey curve displays the suggestion of a standardized force profile (see discussion). To have a better comparison, the *x*- and *y*-axes are standardized (0–10 s and 0–400 N). The ID (Beg_no.) and gender (f = female, m = male) are given per tester. The years of experience for each tester is zero.

**Figure 6 diagnostics-10-00996-f006:**
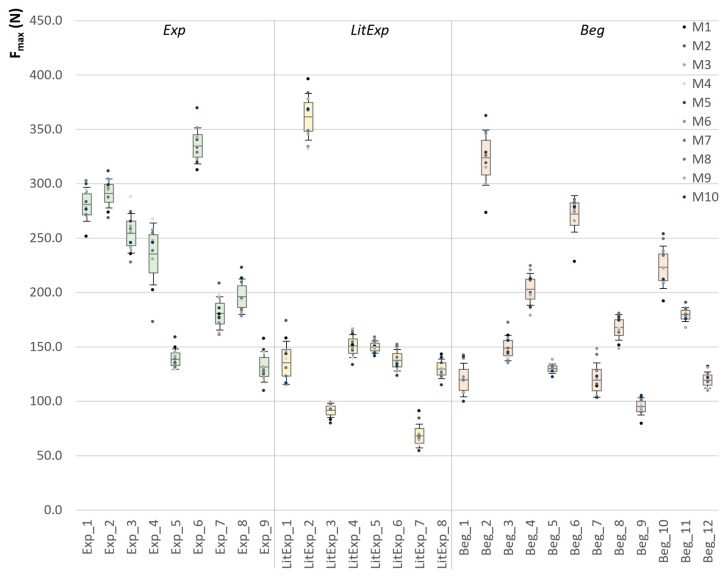
Overview of F_max_ of the repeated trials for each tester. Displayed are the arithmetic means (M), 95%-confidence intervals (CI) and standard deviations (SD; error bars) of the F_max_ (N) of the 10 trials of each tester sorted into the groups experienced (Exp; green, *n* = 9), little experienced (LitExp; yellow, *n* = 8) and beginners (Beg; orange, *n* = 12). The analysis of variance for repeated measurements (ANOVA RM) are significant for Beg (*p* = 0.039) and LitExp (*p* = 0.033), not for Exp (*p* > 0.05).

**Figure 7 diagnostics-10-00996-f007:**
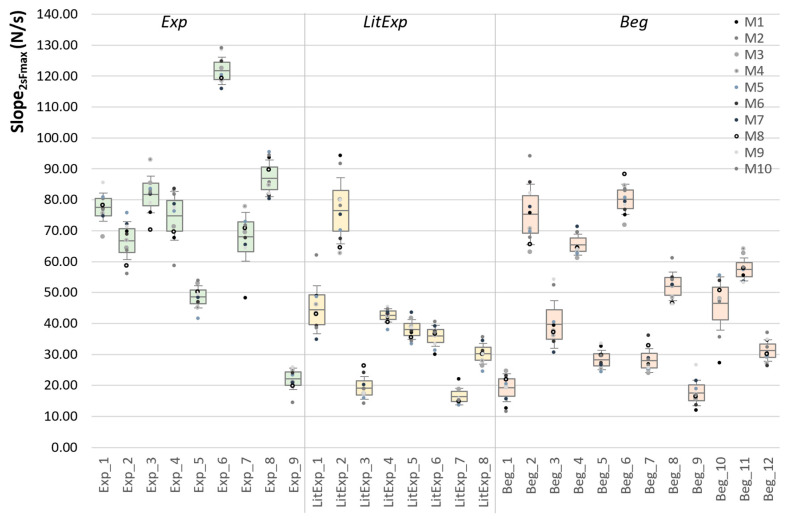
Overall Slope. Displayed are the arithmetic means (M), 95% confidence intervals (CI) and standard deviations (SD; error bars) of the slope_2max (N/s) of the 10 trials of each tester sorted into groups experienced (Exp; green, *n* = 9), little experienced (LitExp; yellow, *n* = 8) and beginners (Beg; orange, *n* = 12).

**Figure 8 diagnostics-10-00996-f008:**
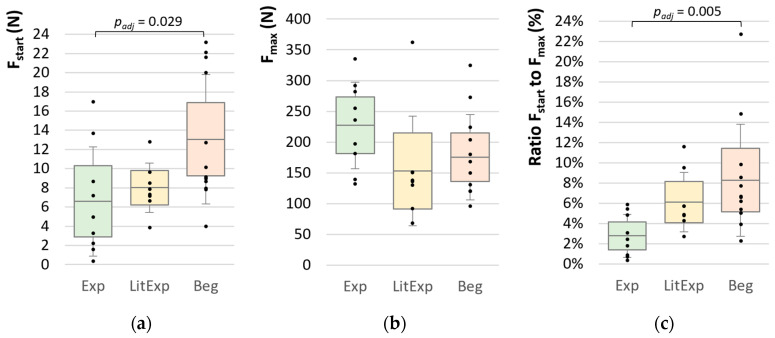
Comparisons of force parameters between the groups. Displayed are the Ms, SDs (error bars) and 95%-CIs of (**a**) the starting force (F_start_ (N)), (**b**) the maximum force (F_max_ (N)) and (**c**) the ratio of F_start_ to F_max_ (%) of the 10 trials in the groups experienced (Exp; green), little experienced (LitExp; yellow) and beginners (Beg; orange). The significant results are displayed by adjusted *p*-values.

**Figure 9 diagnostics-10-00996-f009:**
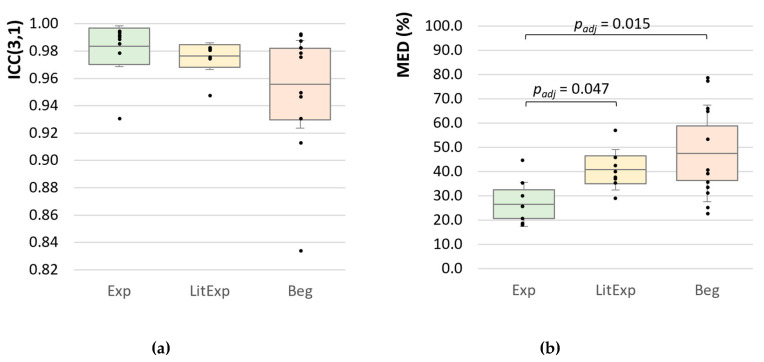
Intraclass correlation coefficient and normalized mean Euclidean distance. Displayed are the Ms, SDs (error bars) and 95%-CIs of (**a**) the ICC(3,1) and (**b**) the MED (%) comparing the groups experienced (Exp; green), little experienced (LitExp; yellow) and beginners (Beg; orange). For the significant comparisons adjusted *p*-values are given.

**Figure 10 diagnostics-10-00996-f010:**
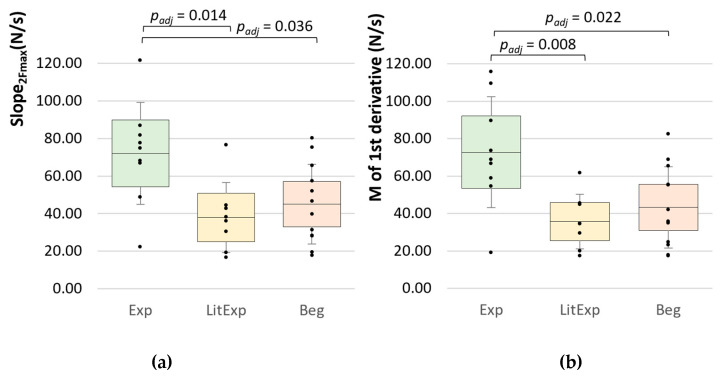
Inter-group comparisons of the overall slope parameters. Displayed are the Ms, SDs (error bars) and 95%-CIs of (**a**) the averaged slope from start to F_max_ (Slope_2max) (N/s) and (**b**) the averaged 1st derivative of the 10 force profiles (N/s) comparing the groups Exp (green), LitExp (yellow) and Beg (orange). Significant comparisons are marked with p_adj_-values.

**Figure 11 diagnostics-10-00996-f011:**
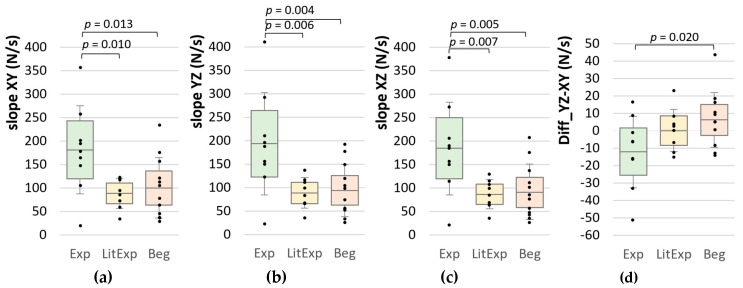
Slope parameters in the linear section XYZ. Displayed are the Ms, SDs (error bars) and 95%-CIs of the slope in the linear part of (**a**) between XY, (**b**) between YZ, (**c**) between XZ and (**d**) the difference between YZ and XY comparing the groups Exp (green), LitExp (yellow) and Beg (orange). The significant pairwise comparisons are marked with *p*-values (not adjusted).

**Figure 12 diagnostics-10-00996-f012:**
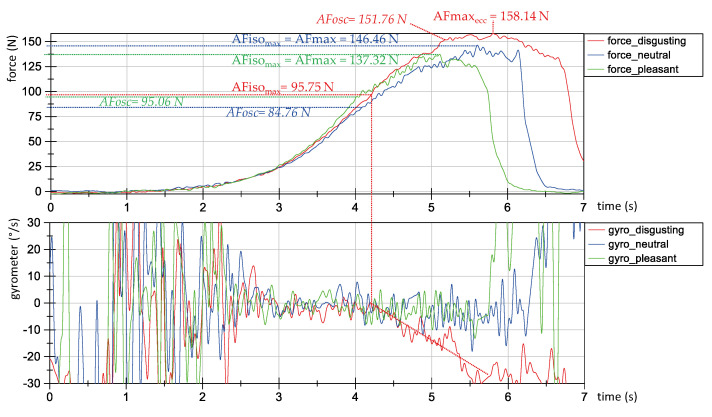
Oscillations during MMT measurements in interaction of tester and participant. Exemplary force (above) and gyrometer signals (below) during MMT measurements using the handheld device. An experienced tester performed the MMT of the hip flexors of one female participant during disgusting (red), neutral (blue) and pleasant (green) odors. The force values (N) for the parameters maximum Adaptive Force (AF_max_), maximum isometric Adaptive Force (AFiso_max_) and the Adaptive Force in the moment of oscillation onset (AF_osc_) are given.

**Figure 13 diagnostics-10-00996-f013:**
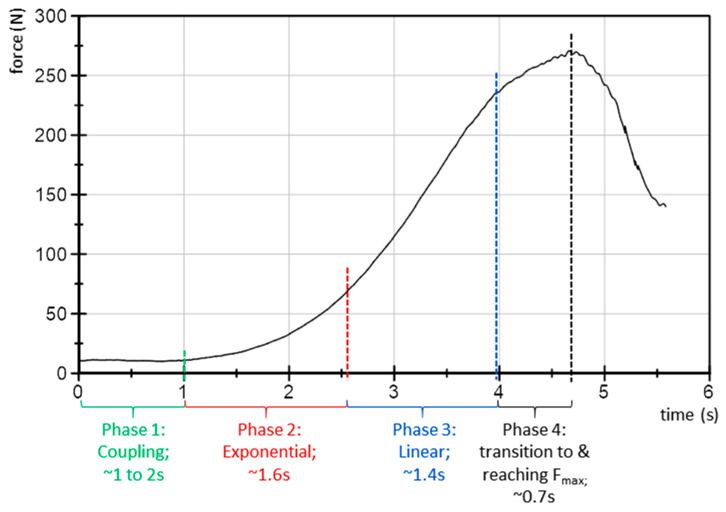
Suggestion for a standardized force profile including the proposed four phases considering the neurophysiological characteristics of forward controlling.

**Figure 14 diagnostics-10-00996-f014:**
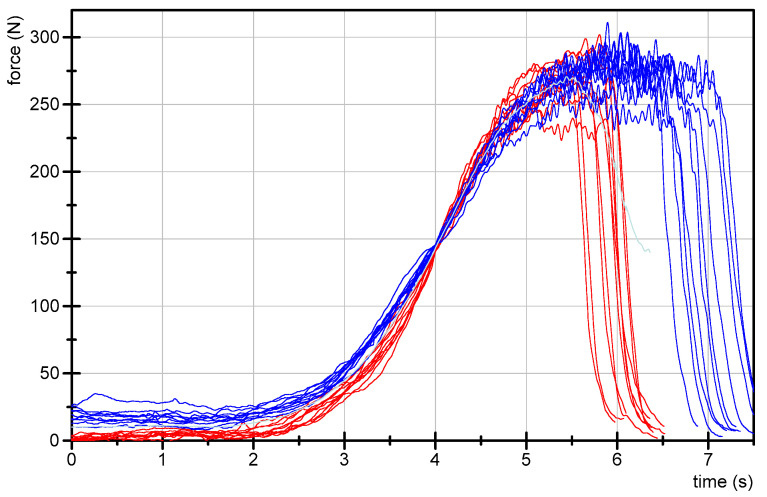
Ten repeated force curves of tester 1 (female, red) and tester 2 (male, blue) against a stable resistance in the MMT setting of the hip flexors with fixed leg (filtered with Butterworth, cut-off frequency 20 Hz, filter degree 5). The grey curve illustrates the average curve of all 20 trials, which is suggested as standardized force profile.

**Table 1 diagnostics-10-00996-t001:** Anthropometric data and years of test experience (MMT exp.) sorted by groups into experienced (Exp), little experienced (LitExp) and beginners (Beg).

Group	*n*	Gender		Age (yrs.)	BMI (kg/m^2^)		MMT exp. (yrs.)
		Male	Female		Male	Female	
Exp	9	7	2	38.33 ± 11.5	27.1 ± 2.7	25.8 ± 7.4	9.22 ± 7.56
LitExp	8	3	5	34.00 ± 8.55	22.8 ± 2.4	22.4 ± 3.4	1.38 ± 0.92
Beg	12	3	9	25.58 ± 3.94	21.4 ± 1.3	22.8 ± 2.2	0.00 ± 0.00

**Table 2 diagnostics-10-00996-t002:** Intraindividual slope in the linear section. Results of statistical tests (RM ANOVA for Exp and LitExp; Friedman test for Beg) analyzing the slope in the linear section XYZ between the 10 repetitions within each group Exp, LitExp and Beg, respectively. Given are the test statistics F or *χ*^2^, significance *p* and effect size η^2^ for RM ANOVA.

Slope Section	Exp			LitExp			Beg	
	F(9,72)	*p*	η^2^	F(9,63)	*p*	η^2^	*χ*^2^(9)	*p*
XY	1.322	0.241	0.142	1.369	0.280	0.164	3.643	0.933
YZ	1.777	0.087	0.182	0.271	0.837	0.037	7.745	0.530
XZ	1.829	0.078	0.186	0.547	0.727	0.073	4.641	0.864
Diff_YZ-XY	0.838	0.584	0.095	0.672	0.581	0.088	5.636	0.776

**Table 3 diagnostics-10-00996-t003:** Inter-group comparisons of force parameters, normalized mean Euclidean distance (MED) and intraclass correlation coefficient (ICC(3,1)). Arithmetic means (M), standard deviations (SD), coefficients of variation (CV) and statistical results of Kruskal–Wallis test (Chi-square (*χ*^2^), significance *p*) for group comparisons between experienced (Exp), little experienced (LitExp) and beginners (Beg) of the parameters starting force (M_F_start_, (N)), maximum force (M_F_max_, (N)), ratio of F_start_ to F_max_ (%), ICC(3,1) and MED (%). For significant results, the pairwise comparisons of Dunn–Bonferroni post-hoc test are given by z-values, adjusted *p*-values (p_adj_) and effect sizes r.

Para-Meter	Group	M	SD	CV	Χ^2^	*p*	Dunn-Bonferroni Post-hoc Test			
							Value	Exp vs. LitExp	Exp vs. Beg	LitExp vs. Beg
M_F_start_ (N)	Exp	6.59	5.69	0.86	7.282	**0.026**	z	−0.672	−2.583	−1.780
	LitExp	8.01	2.57	0.32			p_adj_	1.000	**0.029**	0.225
	Beg	13.05	6.75	0.52			r	-	**0.564**	-
M_F_max_ (N)	Exp	227.17	70.26	0.31	5.338	0.069				
	LitExp	153.13	89.09	0.58						
	Beg	175.26	69.53	0.40						
Ratio F_start_ to F_max_ (%)	Exp	2.79	2.12	0.758	9.992	**0.007**	z	−1.960	−3.337	−0.943
	LitExp	6.12	2.93	0.479			p_adj_	**0.150**	**0.005**	**1.000**
	Beg	8.29	5.55	0.67			r	-	**0.684**	-
ICC(3,1)	Exp	0.984	0.02	0.021	6.424	**0.040**	z	2.002	2.367	0.156
	LitExp	0.976	0.012	0.012			p_adj_	0.136	0.054	1.000
	Beg	0.956	0.046	0.048			r	-	0.517	-
MED (%)	Exp	26.50	9.10	0.343	9.122	**0.010**	z	−2.414	−2.804	−0.139
	LitExp	40.72	8.30	0.204			p_adj_	**0.047**	**0.015**	1.000
	Beg	47.51	19.92	0.419			r	**0.585**	**0.612**	-

Significant results with *p* < 0.05 are written in bold.

**Table 4 diagnostics-10-00996-t004:** Distribution of testers (*n*) according to the amount of ICC(3,1) sorted by groups experienced (Exp), little experienced (LitExp) and beginners (Beg).

ICC(3,1)	Exp	LitExp	Beg
0.989–1	6	0	2
0.979–0.988	2	5	4
<0.979	1	3	6

**Table 5 diagnostics-10-00996-t005:** Distribution of testers (*n*) according to the amount of MED sorted by groups experienced (Exp), little experienced (LitExp) and beginners (Beg).

MED	Exp	LitExp	Beg
<26%	6	0	2
0.26–0.40	2	4	4
>0.40	1	4	6

**Table 6 diagnostics-10-00996-t006:** Arithmetic means (M), standard deviations (SD), coefficient of variations (CV) and statistical results (F value, significance *p*, eta square (η^2^), effect size d and adjusted *p*-value (p_adj_) using Bonferroni post-hoc test) of the MANOVA of slope parameters in comparing the groups experienced (Exp), little experienced (LitExp) and beginners (Beg).

Parameters	Groups	M	SD	CV	F	*p*	η^2^	d	Bonferroni Post-hoc Test *p_adj_*		
									Exp vs. LitExp	Exp vs. Beg	LitExp vs. Beg
Slope_2max (N/s)	Exp	72.03	27.16	0.377	5.635	**0.009**	0.302	0.317	**0.014**	**0.036**	1.000
	LitExp	37.92	18.62	0.491							
	Beg	45.06	21.27	0.472							
M 1st derivative (N/s)	Exp	72.785	29.551	0.406	6.516	**0.005**	0.334	0.350	**0.008**	**0.022**	1.000
	LitExp	35.697	14.574	0.408							
	Beg	43.313	21.714	0.501							
Slope_XY (N/s)	Exp	181.33	94.13	0.519	4.870	**0.016**	0.273	0.283	**0.031**	**0.039**	1.000
	LitExp	88.68	31.90	0.360							
	Beg	100.15	64.37	0.643							
Slope_YZ (N/s)	Exp	193.35	108.86	0.540	6.124	**0.007**	0.320	0.338	**0.019**	**0.013**	1.000
	LitExp	88.58	32.71	0.369							
	Beg	93.93	55.53	0.591							
Slope_XZ (N/s)	Exp	185.08	99.925	0.540	6.039	**0.007**	0.317	0.334	**0.020**	**0.014**	1.000
	LitExp	86.82	31.02	0.357							
	Beg	90.88	57.40	0.632							
DiffSlope YZ-XY (N/s)	Exp	−12.03	20.73	−1.724	3.129	0.061	0.194	0.198	0.436	0.059	1.000
	LitExp	0.099	12.23	123.97							
	Beg	6.216	15.652	2.518							

Significant results with *p* < 0.05 are written in bold.

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
