# Peer review of "Manual Muscle Testing—Force Profiles and Their Reproducibility"

_diagnostics, 2020, doi:10.3390/diagnostics10120996_

Round 1

Reviewer 1 Report

The purpose of this work is to shed light on criticism associated to the manual muscle testing (MMT) and to propose a suitable suggestion to improve this diagnostic tool.

Regarding the introduction, I found the structure of the text flowing and understandable.

The authors clearly sum up the state of the art, recent papers are also reported and MMT criticisms are properly described.

The results obtained are clearly analyzed and discussed taking into account weak points of the study (i.e. the small sample size in the group).

The standardized assessment proposed by authors and supported by the results obtained could be a suitable starting point to improve the reproducibility and robustness of this diagnostic tool.

Author Response

Dear Reviewer 1,

thank you for the positive review – we are happy to meet your requirements!

According to your review, we have made no changes.

Best regards

the authors

Reviewer 2 Report

The submitted manuscript is a work that addresses current and necessary topics. I have only a few minor recommendations and remarks about the manuscript:
In the Introduction, I lack information about other authors who have developed manual testing, in addition to the cited Kendall and Lowett (eg Janda and others) - I recommend adding links to other authors. In connection with manual testing, I would like to point out that it is not possible to mix the principles of testing according to Kendall and according to Goodheart (Applied Kinesiology), as these are different procedures, it can be said with a different goal.
In connection with the theoretical information on manual testing, the areas of use are well described, but I lack the indication of areas where manual testing is not used or is not appropriate.
Material and Methods are clearly described, the chosen procedures are transparent as well as the performed analyzes.
The results are also well described, the graphs are illustrative and clear. However, it should be noted that the testing method used in the present work is not the procedure that would be used in routine manual testing according to Kendall et al., Since isometric contraction is not used there. Therefore, it is not entirely possible to transfer the results obtained in the present experiment to all manual tests in general. I recommend mentioning this fact in the conclusion.
After minor modifications, I recommend the work for publication, it is a quality experiment that will contribute resp. will be another small step towards objectifying manual testing methods.

Author Response

Dear Reviewer,

thank you for the constructive criticism. We hope to meet your requirements and suggestions by the following changes and comments:

Point 1: In the Introduction, I lack information about other authors who have developed manual testing, in addition to the cited Kendall and Lowett (eg Janda and others) - I recommend adding links to other authors.

Response 1: We added further information also concerning Janda and Daniels, Williams & Worthingham in the introduction (p. 2, lines 68-85):

“Kendall et al. suggested a 6-degree scale ranging from palsy (“Gone – no contraction felt” [9] (p. 23)) up to full power (“normal – muscle can hold the test position against strong pressure” [9] (p. 20)). This scaling seems to be in accordance with Janda who also supposed a 6-degree rating scale, ranging from “no evidence of contractility” (Grade 0) to “normal” (Grade 5), whereby Grade 5 is characterized by a “very strong muscle with a full range of movement and able to overcome a considerable resistance.” [10] (p. 2). Both approaches differentiate between a muscle strength and a muscle length test [9,10]. But in contrast to Kendall et al., Janda´s approach – based on the works of Daniels, Williams and Worthingham [11] – is aimed on a concentric contraction of the tested muscle (partly against external resistance provided by the examiner). Because the present study is focused on the so-called “break test” (see below), we further refer to Kendall et al. The muscle strength test includes the determination of the patient’s “strength of the muscle holding in the test position (…) against the examiner’s pressure” [9] (p. 16). This is in accordance with the MMT after Goodheart, as it is utilized in Applied Kinesiology, which simply distinguishes between two states, a stable one (comparable with degree “normal” after Kendall et al. [9] or rather degree 10 appropriate to the IMACS manual [8]) and a state of instability (comparable with degrees “gone” or 10, respectively).

Comparing different MMT techniques – referring to the “strength test” of Kendall [9] or the MMT of Goodheart [12] –, Conable and Rosner gave an overview of the kind of action between subject and tester [13].”

Point 2: In connection with manual testing, I would like to point out that it is not possible to mix the principles of testing according to Kendall and according to Goodheart (Applied Kinesiology), as these are different procedures, it can be said with a different goal.

Response 2: We agree with you regarding the lower grades after Kendall (“Zero” up to “Fair”) which are mostly dealing with true pathologies (like paresis and others). From our point of view (and experiences) and according to the literature, concerning the upper grades the “strength test” according to Kendall is quite comparable to the test according to Goodheart. Both tests pursue the goal to test the resistance of the patient’s muscle against a pressure, which is applied by the examiner. Both approaches distinguish between a full stable resistance without release (“Normal” after Kendall) and a state characterized by yielding despite of a significant resistance (after Kendall the grades immediately below “Normal” referred to as “Good” or “Fair+”, respectively). We have included some information in the paragraph (see 1.). We hope that this is in accordance with your recommendation.

Point 3: In connection with the theoretical information on manual testing, the areas of use are well described, but I lack the indication of areas where manual testing is not used or is not appropriate.

Response 3: The areas where MMT is not used is very large. Since the manuscript already is very long and focuses on the areas where MMT is used, we would prefer to not include further information on this topic. If you assess this as necessary we, of course, would add this information.

Point 4: Material and Methods are clearly described, the chosen procedures are transparent as well as the performed analyzes.

Response 4: Thank you. We are happy to meet your requirements.

Point 5: The results are also well described, the graphs are illustrative and clear. However, it should be noted that the testing method used in the present work is not the procedure that would be used in routine manual testing according to Kendall et al., Since isometric contraction is not used there. Therefore, it is not entirely possible to transfer the results obtained in the present experiment to all manual tests in general. I recommend mentioning this fact in the conclusion.

Response 5:

  • According to Kendall et al (2005), the procedure of “muscle strength” test is performed with an isometric contraction: “strength of the muscle holding in the test position (…) against the examiner’s pressure” (Kendall et al., 2005) (p. 16). The test after Goodheart runs this way as well.
    Nevertheless, we are completely with you. The results obtained in the investigation of course cannot be directly transferred to routine MMT, since the patient’s leg was fixed, in order to rule out the uncertainties coming from the patient´s leg. The aim was to investigate the reproducibility of the examiner´s part. This is the first prerequisite of sufficient testing.
  • There are some passages in the text where this problem is addressed, e.g. p. 2 lines 77, 90; p. 21 lines 31ff, p. 22 line 666; p. 22 line 681; p. 26 lines 783f; We hope thereby to meet your requirements.

Thank you for your effort!

Best regards,

the authors

Reviewer 3 Report

Bittman et al perform an investigation into the variability of manual muscle testing which is an important consideration for a test that is considerably subjective. The study is valuable to the field and is in the most part well designed, I have a few minor comments that I feel need to be addressed.

In Figures 6 and 7 can the authors comment on the fact that the 95% CI for the expert testers seem to be wider than those for the little experience testers. 

For the figures it might be helpful to plot the actual data points in addition to the box plots.

The discussion is very lengthy and would benefit from being more concise with a clear statement of the significance of the findings for the future use of the diagnostic test.

Author Response

Point 1: In Figures 6 and 7 can the authors comment on the fact that the 95% CI for the expert testers seem to be wider than those for the little experience testers. 

Response 1:

  • In general, the CI tend to be wider with higher Fmax. We calculated the Pearson’s correlation coefficient r for Fmax and the individual CI-range of Fmax for each participant. Considering the whole group of Exp, LitExp and Beg together, r = 637 (p < 0.001). Looking into the single groups of Exp, LitExp and Beg the correlation is not significant anymore, e.g. Exp with r = 0.383, p = 0.308. The same was performed for the M of slope and the CI-range of slope: r = -0.106, p = 0.583. It remains open, why this characteristic appears. Nevertheless, there are individual characteristics in single examiners like a wide CI in the experienced Exp_4 or a very good reproducibility with a narrow CI of Fmax in the very beginner Beg_11.
  • We wondered were we could include this information and considered it as too extensive. We hope the information given in discussion p. 19, lines 522-528, will be accepted.

Point 2: For the figures it might be helpful to plot the actual data points in addition to the box plots.

Response 2: We added the data point to all plots with boxes (Fig. 6 to 12). Thank you for the suggestion, this really gives further information and improves the paper.

Point 3: The discussion is very lengthy and would benefit from being more concise with a clear statement of the significance of the findings for the future use of the diagnostic test.

Response 3: Indeed, the discussion seems to be broader than necessary regarding the actual central topic. We did a revision trying to reach more conciseness, especially in the discussion (deleted second paragraph (p. 19 in re-submitted version), moved some data of 4.1. into results section and shorten especially 4.1.). However, this seems to be rather minor cancellations. We found it valuable to add the neurophysiological considerations to underpin our suggested model of a force profile. For this the aspects of the central processing rhythms, the forward control and the oscillations seem to be important aspects. On the other hand, considering the length of the paper your criticism is understandable to us. In case you insist on shortening the text, we could remove the sections about the oscillations. This would be done in several sections.